

# Exploring extreme event attribution by using long-running meteorological observations

Erik Holmgren[1,2] and Erik Kjellström[1,3]

[1]Rossby Centre, Swedish Meteorological and Hydrological Institute, Norrköping, Sweden
[2]Division of Geoscience and Remote Sensing, Department of Space, Earth and Environment, Chalmers University of Technology, Gothenburg, Sweden
[3]Department of Meteorology and Bolin Centre for climate research, Stockholm University, Stockholm, Sweden

**Correspondence:** Erik Holmgren (erik.holmgren@chalmers.se)

**Abstract.** Despite a growing interest in extreme event attribution, attributing individual weather events remains difficult and uncertain. We have explored extreme event attribution by comparing a widely adopted method for probabilistic extreme event attribution to a more analogue approach utilising the extensive, and long-running, network of meteorological observations available in Sweden. The long observational records enabled us to calculate the change in probability for two recent extreme

events in Sweden without relying on the correlation to the global mean surface temperature, as is usually done in the reference method. Our results indicate that the two methods generally agree on the sign of attribution for an event based on daily maximum temperatures. However, the reference method results in a weaker indication of attribution compared to the observations, where 12 out of 15 stations indicate a stronger attribution than found by the reference method. On the other hand, for a recent extreme precipitation event, the reference method results in a stronger indication of attribution compared to the observations.

For this event, only two out of ten stations exhibited results similar to the reference method.

## 1   Introduction

Anthropogenic greenhouse gases are the main drivers of the observed increases in global temperatures during the 20th century (IPCC, 2021; Eyring et al., 2021). Even though the global warming is accompanied by a notable increase in the intensity, and frequency, of local extreme temperature and precipitation events (Trenberth, 2011; Seneviratne et al., 2021), linking individual

extreme weather events to anthropogenic emissions remains a challenge.

Extreme weather events typically display unusual meteorological properties, cause severe effects on society, or occur relatively infrequently. However, the frequency and intensity of many of today's extreme events are expected to change with the ongoing changes of the global climate. For some types of events, changes in their climatology have already been observed (e.g. Holland and Bruyère, 2014; Wilcke et al., 2020). Extreme weather, and its consequences, have often already been experienced,

which makes it particularly interesting. A popular question to ask is if any specific, especially intense, weather event was caused by changes in the climate, and more specifically anthropogenic changes.

The relatively novel field of extreme event attribution (EEA) arose out of the need to try to answer questions like this. EEA is a collection of methods used to investigate if an extreme event can be attributed to any one forcing, such as anthropogenic




climate change (e.g. Stott et al., 2016; van Oldenborgh et al., 2021). There are several approaches to EEA, where two of the

more common ones are the risk-based approach and the storyline approach. In the risk-based approach, as described in e.g. Stott et al. (2016), the question of attribution is framed as probabilistic: *How has forcing x changed the likelihood of event y?* Here, it is the change in risk that an event occurs that is attributed to the changed forcing, rather than the event itself. This circumvents the difficult question of investigating the causal relationships of an extreme weather event. The storyline approach (e.g. Hoerling et al., 2013) instead focuses on the underlying physical processes in combination with the stochastic nature

of an event. It tries to quantify the role of natural variability and forcings, such as increased greenhouse gases, sea surface temperature (SST) and soil moisture, on the event. Due to these differences, EEA studies conducted on the same event, e.g. the Russian Heatwave in 2010, employing different methods, can appear to have reached contrasting conclusions (e.g. Dole et al., 2011; Rahmstorf and Coumou, 2011), even if both studies turned out to be compatible (Otto et al., 2012). Similarly, the use of different datasets can affect the outcome of an attribution study.

Data representing the pre-industrial reference period is scarce and often not available. Instead, it is possible to make use of the global mean surface temperature (GMST) to shift, or scale, a distribution of the variable describing the event (See 2.1), and in this way represent the pre-industrial climate. These relationships are generally well-defined at global scales. Regionally, however, there are many factors influencing how changes in the global climate propagate and affect the local climate (Doblas-Reyes et al., 2021).

The last decade has seen a rapid increase in both the number of publications and general interest of EEA studies. A notable example is the BAMS special issue *Explaining Extreme Events* (e.g. Herring et al., 2022), which has been published annually since 2011. Olsson et al. (2022) argues that the increasing interest in EEA is connected to the ongoing development of the framework for Loss and Damages (L&D), where the attribution of single events could become a useful tool (Parker et al., 2015). The possible use of EEA in future L&D programs, combined with the increasing societal interest in extreme weather

events, makes the exploration and evaluation of the suggested methods both compelling and important.

One particularly interesting aspect of the reference method for probabilistic attribution is the assumption of a linear relationship between GMST and the variable describing the event, and how this is used to represent the pre-industrial climate. However, this linear relationship will likely not capture other factors affecting the local response to global changes. Hence, any local effects will not be included in the representation of the pre-industrial period, since it solely relies on how well the variable

correlates to the GMST. In turn, this could affect the outcome of an attribution study, where results stem from the difference in probability during the pre-industrial period and the recent past.

We aim to explore the proficiency of shifting, and scaling, the climate by GMST in the simplest way possible: by comparing it to observations. To achieve this, we will use the long-running observational network in Sweden to investigate two of the most notable extreme events in Sweden during the recent years: the particularly warm summer of 2018, in this study focused

on southern Sweden, and the heavy precipitation event hitting the Swedish city Gävle in August 2021. The heatwaves during the summer of 2018 have been featured in multiple recent studies (Leach et al., 2020; Yiou et al., 2020; Wilcke et al., 2020). Contrastingly, while the precipitation event in Gävle was heavily featured in the media and has been examined by the Swedish meteorological and hydrological institute (SMHI), studies focusing on the attribution of the event are lacking. By including





the two events in the same study, we also aim to highlight the differences with the attribution of events based on different
meteorological variables (temperature and precipitation).

For both of these events, we will perform two sets of attribution studies, employing different methods, and compare their results. The first analysis is based on the rapid attribution framework from Philip et al. (2020), while the second analysis will instead make use of data from several stations with observational records covering both the current and a pre-industrial period.

## 2 Method

In this study, we will employ parts of the rapid attribution framework from Philip et al. (2020) to investigate the possible attribution of two recent events in Sweden: The warm summer of 2018 and the heavy precipitation event in Gävle on the 17-18th of August 2021. Alongside this more commonly used attribution method, we will also perform an analysis based on long-running series of meteorological observations.

### 2.1 Probabilistic extreme event attribution

The rapid attribution framework from Philip et al. (2020) is a risk-based approach to attribution. It consists of steps outlining the preparations, analysis, and communication of an attribution study. In the following section, we will describe parts of the statistical method outlined in the framework.

The final result of a probabilistic attribution study is the probability ratio (PR)

$$PR = \frac{p_1}{p_0}, \tag{1}$$

or fraction of attributable risk (FAR)

$$FAR = 1 - \frac{p_0}{p_1} = 1 - \frac{1}{PR}, \tag{2}$$

where $p_1$ and $p_0$ are the probabilities for the event in the factual (current climate) and counterfactual (pre-industrial climate) worlds (see Fig. 1). PR and FAR are interchangeable, and which one to use depends on how the results will be presented. PR is interpreted as how many times more likely (or unlikely if <1) an event with the same magnitude has become. FAR instead
describes how large fraction of events of the same magnitude that can be attributed to the changed forcing. For instance, if the PR of an event is 2, the interpretation would be that it has become twice as likely. On the other hand, the interpretation of the corresponding FAR= 0.5 is that half of the occurrences of similar events can be attributed to the changed forcing.

To calculate $p_1$ and $p_0$, ideally long observational datasets and climate model output, which contain periods that represent both the current and pre-industrial climate, should be used. The probability of a class of events, in either of the two periods,
can then be sampled from the continuous density function (CDF) of a theoretical distribution fit to data corresponding to that period (see Fig. 1). In most cases data describing the current climate is readily available, either from observations or models, and retrieving $p_1$ is relatively trivial.

Computing $p_0$ requires data of the pre-industrial period. Unfortunately, continuous observations with good spatial coverage from these times are rare. One option is to use climate models. For instance, General Circulation Models (GCMs) part of



the Coupled Model Intercomparison Project (CMIP, Eyring et al. 2016) have a pre-industrial control run which could be used to represent the pre-industrial climate in attribution studies. A second option is to use fixed forcing GCM runs, with for instance prescribed sea surface temperature. However, the resolution of GCMs is generally too low to properly represent many extreme weather events. The increased resolution of regional climate models (RCMs), for instance members of the Coordinated Regional Climate Downscaling Experiment (CORDEX, Jones et al. 2011) ensemble, does enable the models to better represent

extreme weather events. However, the high resolution runs completed in CORDEX traditionally do not include a pre-industrial control period, and thus only cover a period from the middle of the 20th century and forward.

A third option for the pre-industrial climate, and what is used in this and many other attribution studies, is to shift, or scale, the distribution that represents the current climate. This relies on the assumption that the variable used to describe the event shifts or scales with a forcing that has a known climate change signal and historical record. An example of this is the global

mean surface temperature (GMST), commonly used as a key indicator of climate change (e.g. Gulev et al., 2021).

A theoretical distribution, e.g. the GEV distribution, can be described by three parameters: the location, $\mu$, scale, $\sigma$, and shape, $\xi$. To shift a distribution according to its relationship with GMST, $\mu$ is shifted following

$$\mu = \mu_0 + \beta \Delta T. \tag{3}$$

Here $\beta$ is the coefficient of the linear regression between the variable and GMST, and $\Delta T$ is the change in GMST between the

current and pre-industrial period. $\sigma$ and $\xi$ are left unchanged. If the variable is instead assumed to scale with GMST, which is the case for precipitation, $\mu$ and $\sigma$ are changed following

$$\mu = \mu_0 \exp(\beta \Delta T / \mu_0) \tag{4}$$

and

$$\sigma = \sigma_0 \exp(\beta \Delta T / \sigma_0). \tag{5}$$

Either of these approaches will result in a distribution that represents the pre-industrial climate, where the CDF can be sampled to retrieve $p_0$ (see Fig. 1).

In this study, we defined two domains used for the analysis of the events, one for the heat wave in the summer of 2018 and one for the heavy precipitation event in Gävle 2021 (Fig. 2). The summer 2018 domain covers the mainland of Sweden south of 60°N. For the Gävle event we used a region between 59°N and 63°N, and east of 13.5°E. We extracted 30 years of daily

data for the two events, forming the reference periods used to retrieve $p_1$. For the summer of 2018 event, this consisted of daily maximum temperatures between 1989 and 2018, while for the Gävle event, daily precipitation flux between 1991 and 2021 was analysed.

We used the following gridded datasets: GridClim (Andersson et al., 2021), PTHBV (only for Gävle) (Johansson and Chen, 2005; Alexandersson, 2003; Johansson, 2000; Johansson and Chen, 2003), E-OBS (Cornes et al., 2018), and ERA5 (Hersbach

et al., 2020). Additionally, we also used a bias adjusted (Berg et al., 2022), 66-member, version of the Euro-CORDEX ensemble (Coppola et al., 2021; Jacob et al., 2014), as described in Kjellström et al. (2022). GridClim, E-OBS, ERA5, and the CORDEX





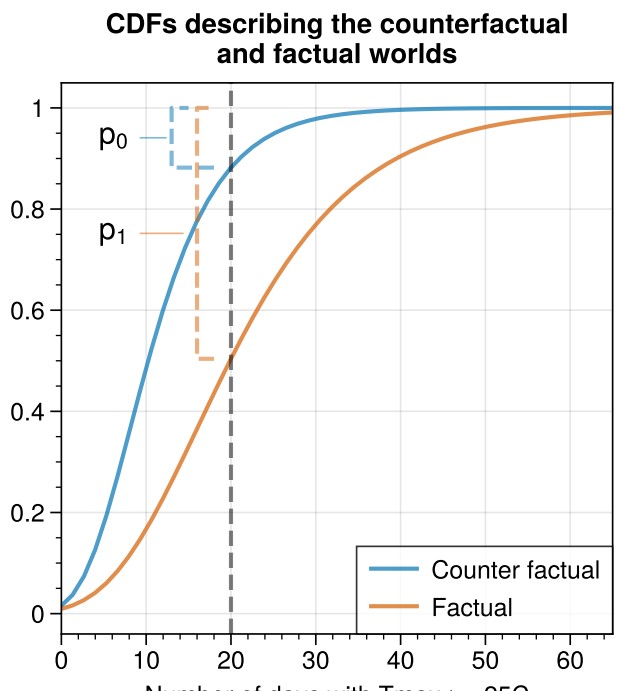

**Figure 1.** Conceptual image describing the relationship between CDF and probability. Here, the two CDFs describe the distributions of the annual number of days with Tmax $\geq 25°$C in a factual and counterfactual world. An event threshold of 20 days is indicated by the grey vertical line. The corresponding event probabilities $p_1$ and $p_0$ are visualised as bracketed annotations.

ensemble all provided data until the end of 2018, while PTHBV covered up until the end of 2021. To assess how well the individual members of the CORDEX ensemble represented observations, we computed the same four metrics as used in Bayerisches Landesamt für Umwelt (2020), between 1989 and 2018 over Sweden, using GridClim as the reference dataset.

For all datasets grid points outside the Swedish mainland were masked. Following this, we calculated climate indicators, further described in section 2.3. For the index describing the summer 2018 event, we calculated the domain average for each year. Since heavy precipitation events are generally more localised compared to heat waves, we instead opted to calculate the annual domain maxima for the Gävle 2021 event.

Following Philip et al. (2020), we computed the linear regression between the 4-year rolling mean GMST (Hansen et al.,
2010) and the annual time series of each index. For the CORDEX ensemble, the regression coefficient of each ensemble member was used as an additional quality control, where we removed any member with a regression coefficient outside the 95% confidence interval of the regression in the reference dataset (GridClim). For all datasets, we used the regression coefficients to detrend the index-series of the current climate. For each index series, we then fit and evaluated a number of common extreme value distributions and selected one for further analysis (see sec. 2.4). We then used the regression coefficients ($\beta$) to
respectively shift and scale the index distributions describing the summer of 2018 and Gävle 2021 events according to eq. 3, 4





and 5. For each dataset, the distribution of the current climate and the pre-industrial (shifted/scaled) distribution formed a pair from which $p_1$ and $p_0$ could be retrieved and used to calculate FAR/PR (Eq. 1 & 2). The threshold used for the summer of 2018 event was based on the 2018 domain average txge25 in the GridClim product, whereas we used the 2021 domain maximum rx1day in PTHBV for the Gävle 2021 event. For the gridded observations (GridClim, PTHBV, E-OBS, ERA5) we calculated

confidence intervals with a bootstrap of randomly re-sampling the 30-year index-series and performing the previous steps 1000 times. For the CORDEX data, instead of bootstrapping the confidence intervals, FAR from each ensemble member was used to form the distribution from which the confidence intervals could be retrieved.

## 2.2   Attribution using observations

As an alternative to the more common method outlined above, we performed an attribution analysis employing several stations

with long observational records of daily data. To begin, we employed a set of station merges commonly used at SMHI to extend and fill the gaps in the observational records for temperature and precipitation (e.g. Joelsson et al., 2022). This merges nearby stations which are assumed to be representative of the same geographic location but have different temporal coverage. The merged stations were used to select two sets of stations, one for each of the investigated events. The observational records were checked for missing values and a station missing $\geq 15\%$ of the days in the investigated period, during at least one year, were

flagged in the subsequent analysis.

For each event, we selected all stations located inside the domain on the Swedish mainland and the island Gotland (Fig. 2). For the station data we calculated the same climate indices as was done for the gridded datasets (see 2.3). Following the index calculations, we further refined the station selection by requiring each station to provide a continuous 30-year period for both the historic and current climate. For both events, the historic period was defined as the years 1882 to 1911. This

represents a period largely unaffected by anthropogenic climate change and is relatively well covered in the observational records. The current climate period for the summer of 2018 was defined by the years 1989 to 2018, while for the Gävle 2021 event as the years 1992 to 2021. The selection procedure resulted in 15 stations for the 2018 event and 10 stations for the 2021 event. The locations and names of these stations are shown in figure 2. We checked the station data for stationarity using the Kwiatkowski-Phillips-Schmidt-Shin (KPSS) and Augmented Dickey-Fuller (ADF) tests (See Appendix A).

The two periods representing the historic climate and that of the recent past were then used to calculate PR and FAR following eq. 1 and 2, in the same way as done in the probabilistic event attribution (Section 2.1). Here, the threshold for the summer of 2018 was set to the 2018 station averaged txge25, while the 2021 rx1day from Gävle-Åbyggeby was used as a threshold for the Gävle event. Furthermore, we also compared these results to those obtained by either shifting or scaling the distribution representing the current period for each station.

## 165   2.3   Climate indicators

We used two climate indicators to describe the events: the number of days with maximum temperature $\geq 25°C$ (txge25) for the summer 2018 event, and the maximum 1-day precipitation (rx1day) for the Gävle 2021 event. We calculated the indicators



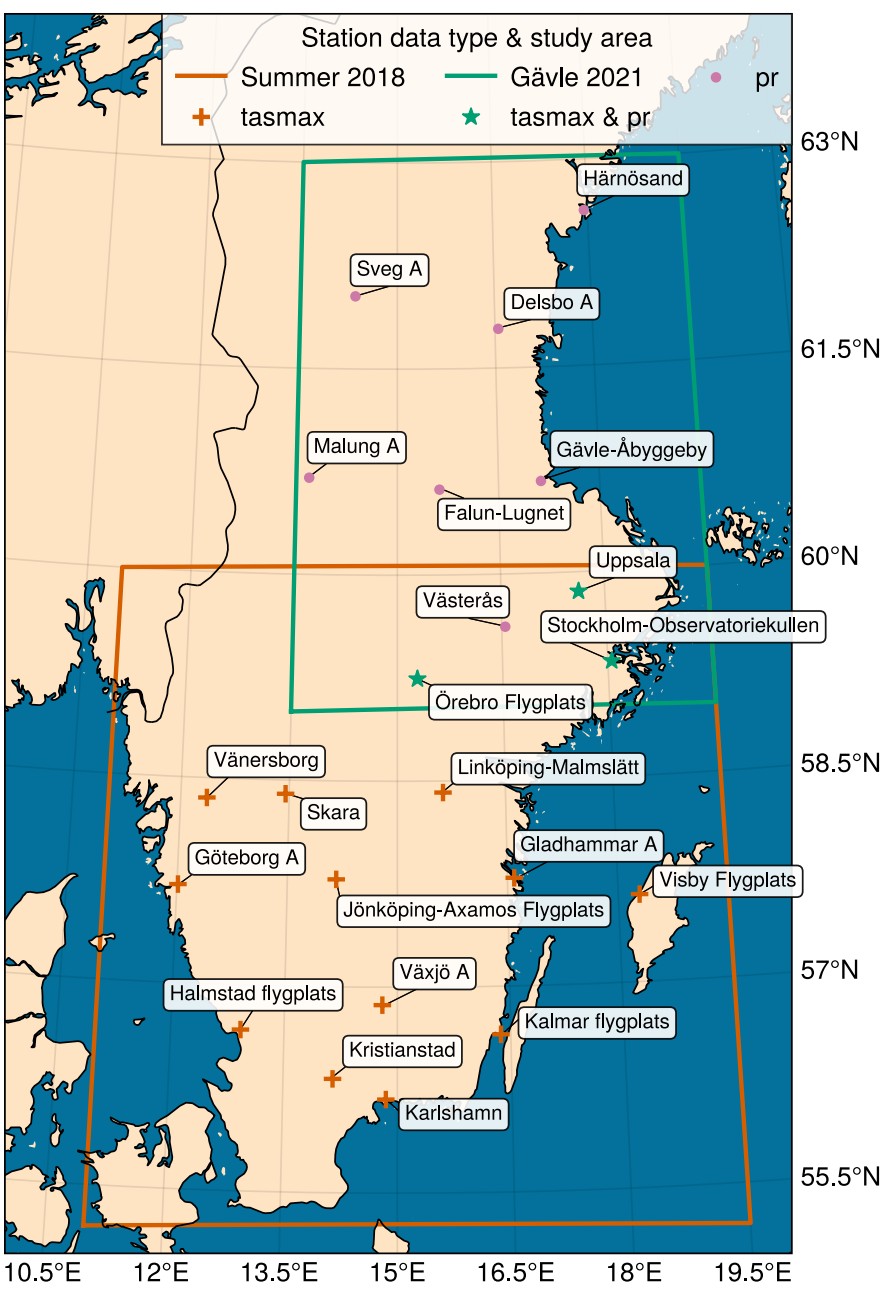

**Figure 2.** Locations of stations employed in this study. Purple dots visualise stations used only for precipitation data, the orange plus are used for temperature data, and green stars show stations used for both precipitation and temperature. Coloured boxes show the outlines for the regions used in selection of the gridded data.





using the software Climix (Zimmermann et al., 2023). For the summer 2018 event, only days within the period from May to August (MJJA) were used to calculate the indices, while rx1day was calculated over the entire year.

## 2.4 A note on distributions

In this study, we used the python package SciPy (Virtanen et al., 2020) to fit, evaluate and sample the distributions used to represent the data. There are multiple distributions suitable to represent extreme distributions, for instance GEV, Gaussian, GPD or Gumbel. We refer to Philip et al. (2020) for further details on the selection of distributions. It is common practice to use a goodness of fit test, such as the Kolmogorov-Smirnov test (KS-test), to evaluate the suitability of the different distributions to represent the data. However, we have found that relying solely on the KS-test for selecting the appropriate distribution insufficient. Most notably, while the GEV distribution tends to show the highest performance in the KS-test, it often results in division by zero errors in Eq. 1 for very high quantiles. The right-skewed Gumbel distribution does not lead to the same division by zero errors, while it still shows good performance in the KS-test. Because of this, we opted to use the right-skewed Gumbel distribution for all probability estimations in this analysis.

## 3 Results and Discussion

### 3.1 Summer of 2018

Almost all the stations employed in the analysis (Fig. 2) recorded $\geq 50$ days with daily maximum temperatures $\geq 25°C$ (summer days) during the summer of 2018 (Fig A4), here defined as May to August (MJJA). There are only a few stations where this is not, by some margin, the highest number of summer days recorded between 1989-2018. Between 1882-1911, there are no years that equals the number of summer days in 2018 among any of the stations (Fig. A3).

Histograms, along with the distributions, generally show a positive difference between the distributions of the current and historic climate for the txge25 index (Fig. 3). Kristianstad, Karlshamn and Linköping-Malmslätt (LM) exhibit the smallest difference between the two periods. The median FAR for LM is $-3.4$ while most other stations exhibit a median FAR $> 0.8$ (Fig. 3). Furthermore, there are no spatial patterns over southernmost Sweden that could explain the negative FAR of LM and Karlshamn (Fig. 4).

The station averaged FAR for the txge25 index has a median of 0.50 with the 5th percentile ($P_5$) $\sim$ -0.78 (Fig. 5). This further highlights LM, where neighbouring stations Skara, Gladhammar A and Örebro Flygplats (see Fig. 2 & 4) display FAR distributions centred $\geq 0.75$. For the adjusted station average (Fig. 5), where Karlshamn, Kristianstad and LM are excluded, median FAR $\sim 0.78$, and $P_5 > 0.1$. The deviating results of the excluded stations are likely not the result of a local response to changes in the climate. Instead, it is more likely a result of the station merging. In some cases, merging implies that the station is moved to a new location. For LM, the station was moved a few kilometres west from central Linköping to the airfield at Malmslätt in 1943. Since temperatures are generally higher in urbanised areas due to the urban heat island (Rizwan et al., 2008), moving the station to a more rural area could introduce erroneous trends to the series (e.g. Tuomenvirta, 2001; Dienst

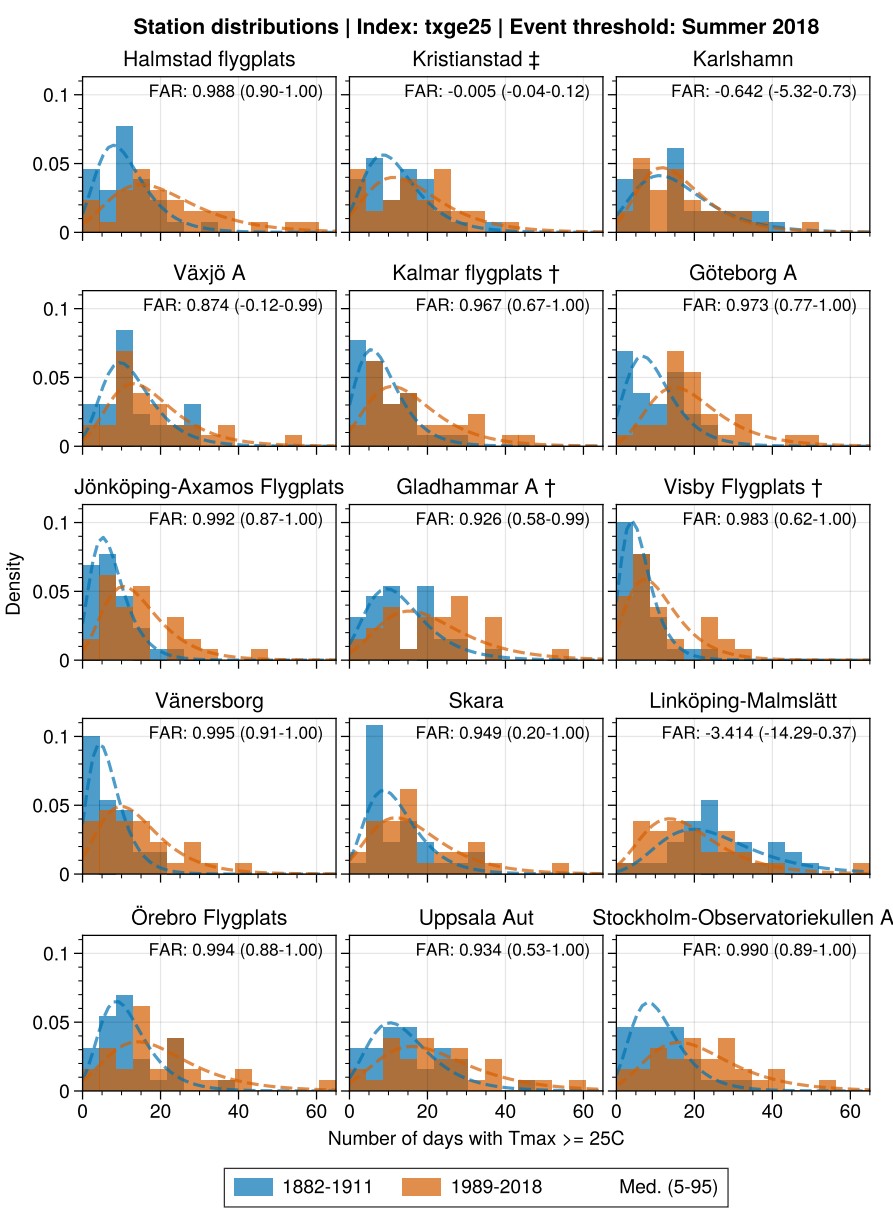

**Figure 3.** Histograms of the txge25 index for periods 1882-1992 and 1989-2018. Dashed lines show the pdf of the Gumbel distribution fit to each period. 2018 is used as a threshold for FAR for each station. † indicate that at least one year miss 15% of the days in the historic period. ‡ is the equivalent for the current period.

et al., 2017). On the other hand, Karlshamn is an example of stations which provides a continuous observational record without merges or changes in location. Here, the implementation of thermometer screens during the 20th century, which generally results in reduced recorded temperatures, could be a part of the explanation.






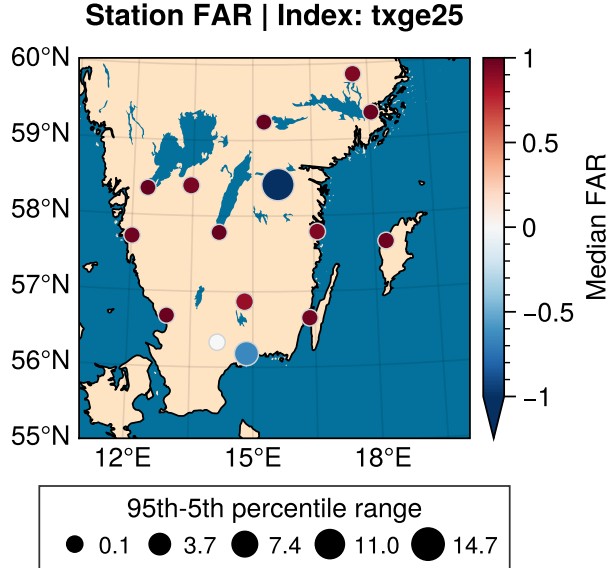

**Figure 4.** Station locations. Marker colour shows the median FAR, while marker size shows the 95th to 5th percentile range.

These are examples of inhomogeneities in the observational record that makes the investigation of trends and climate change difficult. Ideally, when working with signals of climate change in observational data, the data should first be homogenised. Joelsson et al. (2022) presents monthly averages of the 2-metre temperature in Sweden from 1860, based on homogenised data from a high number of stations. The stations used in this study are a subset of the stations used in Joelsson et al. (2022). Since this study makes use of daily temperatures, we can not directly utilise their results. However, their findings can help to further evaluate our results. In general, they found the required temperature corrections to be negative, with larger corrections during summers at the end of the 19th, and beginning of the 20th, century. This implies that the estimated event probabilities during the pre-industrial period in this study are likely too high, which in turn results in an underestimation of FAR.

The analysis of the gridded data products, using shifted distributions, exhibits FAR similar, albeit lower, to that of the station data (Bottom five bars, fig. 5). FAR distributions for GridClim, E-OBS and ERA5 all exhibit a median of 0.4-0.6 and a low spread, with 5th percentiles well above 0. The CORDEX ensemble FAR distribution, here with 62 members, shows a higher median ($\sim 0.8$), however it exhibits a greater spread ($P_5 \sim -4$, $P_{25} \sim 0.2$) compared to the observational based products ($P_5 \sim -0.8$, $P_{25} \sim 0.1$). The weighted average of the gridded products is lower compared to the adjusted station average, but uncertainty ranges overlap.

## 3.2 Gävle 2021

During the event on the 21st of August 2021, the station Gävle-Åbyggeby measured 121 mm of precipitation in 24 hours. This corresponds to the annual maximum one day precipitation (rx1day) at that station in 2021. In 2021, no other station in the study





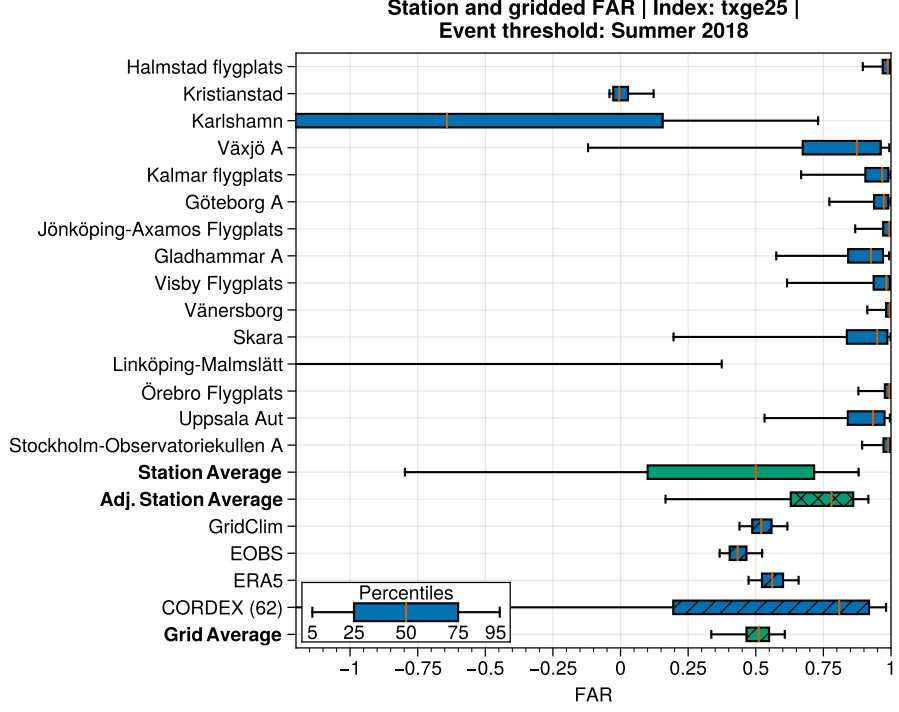

**Figure 5.** FAR synthesis for the summer of 2018, as described by the txge25 index during MJJA. For station data, the factual and counterfactual worlds are represented by observations from 1989-2018 and 1882-1911, respectively. Green bars denote station/dataset averages. Hatched bars display results from gridded data products, whereas the crossed bar shows the adjusted station average. A bar visualise the bootstrap distribution percentiles as described in the inset. Note that the x-axis is limited for increased readability.

area (Fig. 2) recorded a similar amount of precipitation in a single day. However, there are years in the recent past with annual
daily maximums similar to the Gävle 2021 event, both in Gävle and at other stations (Fig. A6). In the historic period there are
only two events, both in Härnösand, with similar magnitudes to the 2021 event (Fig. A5).

For the heavy precipitation event in Gävle 2021, differences between the historic and current climate are small at most of
the 10 stations used to investigate the event (Fig. 6). Some stations (e.g. Gävle-Åbyggeby, Härnösand, Sveg A) exhibit larger
differences, most notably in the tails of the distributions, between the two periods.

The FAR synthesis (Fig. 7) displays the large variability among the stations. Uppsala is the only station where the confidence
interval doesn't include 0, $P_5 \sim 0.2$. Looking at the median, Gävle-Åbyggeby, Sveg A, Uppsala, Västerås, and Örebro D all
exhibit FAR $> 0$. The remaining stations (Falun-Lugnet, Härnösand, Malung A, Stockholm-Observatoriekullen) show a median
FAR $\leq 0$, which indicates that the event has become less likely. These differences are reflected by the large spread exhibited
by the station average in Fig. 7 ($P_{50} \sim -0.2$).

FAR distributions from the gridded analysis of the Gävle 2021 event are shown in the hatched bars in Fig. 7. PTHBV,
GridClim, EOBS and CORDEX exhibit similar medians (0.78-0.98), $P_5 \geq 0.6$. Results from ERA5 does not match the other



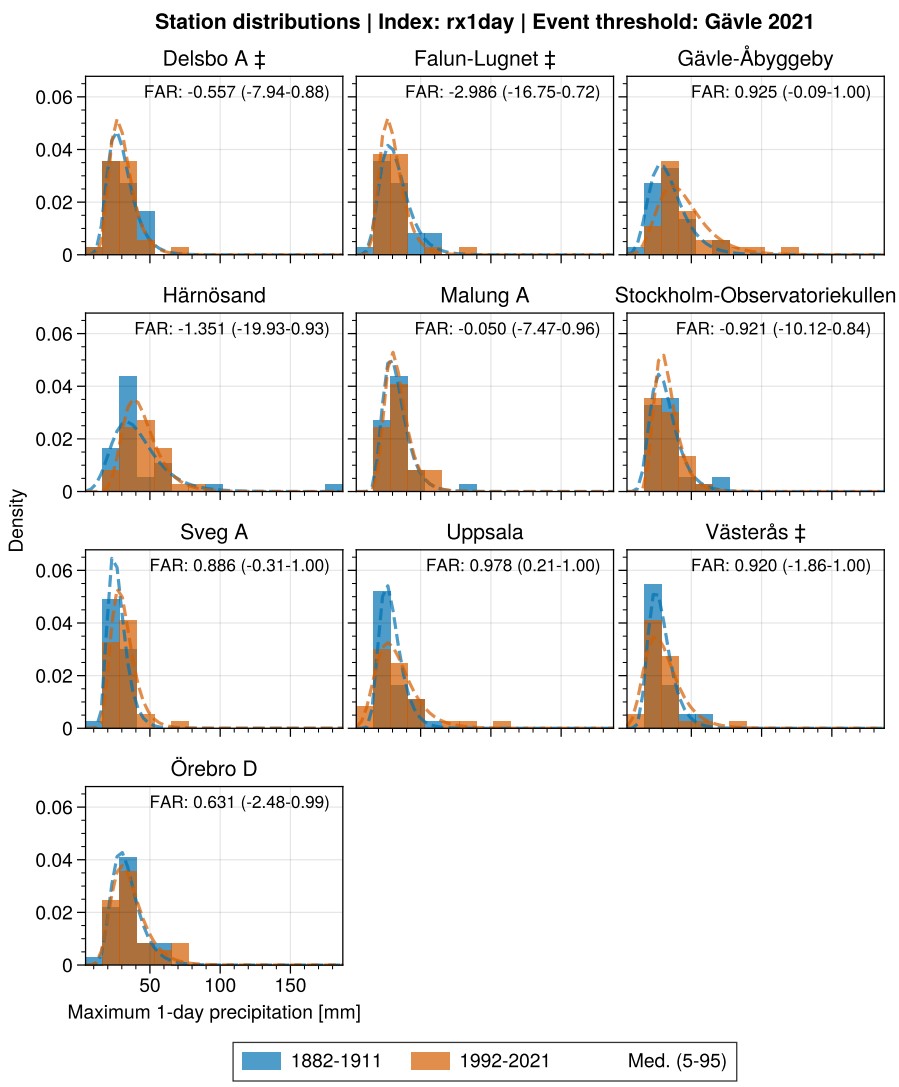

**Figure 6.** Histograms of the annual maximum 1-day precipitation index for periods 1882-1992 and 1992-2021. Dashed lines show the pdf of the gumbel distribution fit to each period. 2018 is used as a threshold for FAR for each station. † indicate that at least one year miss 15% of the days in the historic period. ‡ is the equivalent for the current period.

datasets, with a median FAR $\sim -2.5$ and $P_{95} \sim -2$. We also noted that ERA5 exhibits a negative regression to GMST, as opposed to the other datasets where the regression is generally positive. A contributor to this could be the overall underestimation of rx1day in ERA5 found by Lavers et al. (2022). This requires further investigation, and we chose not to include ERA5 in the

"Grid Average" FAR.

There is some disagreement between the station-based results and the results of the gridded analysis. For stations such as Gävle-Åbyggeby, Sveg A, Uppsala, Västerås, and Örebro D, the median FAR is of the same magnitude as that of the gridded



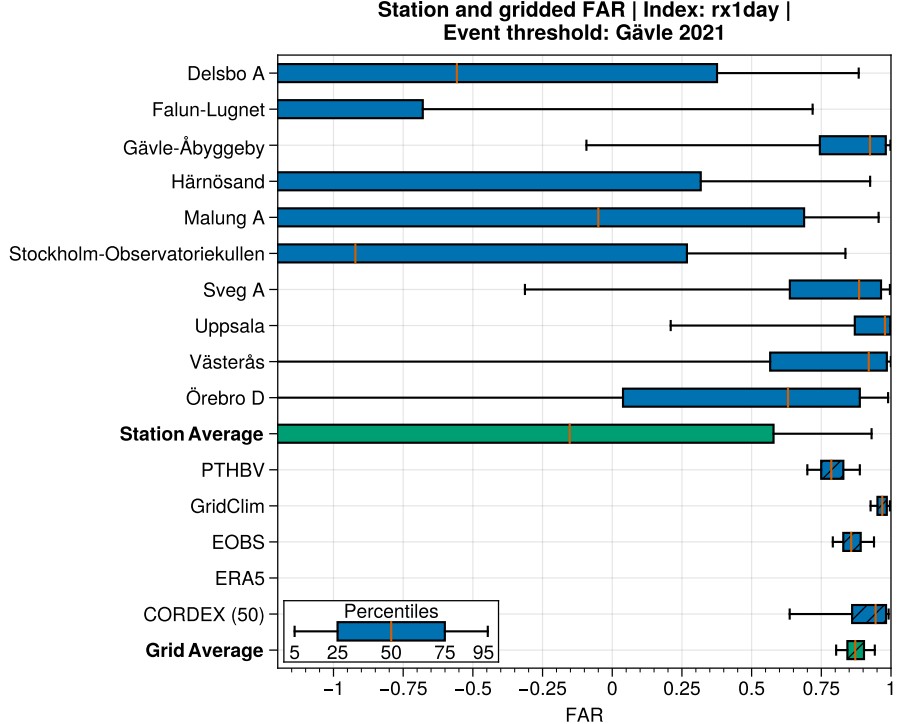

**Figure 7.** FAR synthesis for the heavy precipitation event in Gävle 2021 described by the annual maximum 1-day precipitation (rx1day). The factual and counterfactual worlds are represented by station data from periods 1992-2021 and 1882-1911, respectively. The median station FAR for an event similar to Gävle 2021 is ∼-0.2. Hatched bars display results from gridded data products. A bar visualise the bootstrap distribution percentiles as described in the inset. Note that the x-axis is limited to -1 for readability. The ERA5 FAR distribution lies below this range and is not displayed.

analysis. However, uncertainties are generally greater for the station-based analysis, with lower 5th percentiles. Here it is worth remembering that all FAR distributions, from both gridded data and observations, are based on the same number of samples

and bootstrap iterations. Furthermore, the question of homogeneity in the station data also applies to the precipitation measurements. During the later parts of the 20th century, many stations have been converted from manual to automatic operation in Sweden. Here, the placement of automatic stations were generally more exposed to wind compared to manual stations, where comparisons have shown that automatic precipitation measurements are lower compared to those of manual stations (Alexandersson, 2003).

The two 30-year periods used to represent the pre-industrial (1882-1911) and current (1992-2021) climate were found to be stationary at most of the stations (Fig. A5 & A6). Based on this, we decided not to detrend the station data. This also kept the following analysis closer to the actual observations, removing any dependence on regressions to e.g. GMST. Comparing the





pre-industrial and current climate, most of the stations show distributions with very similar means (Fig. 6). In these cases, the results will be more sensitive to the randomness of the bootstrap, resulting in the large uncertainties for some stations in Fig. 7.

## 3.3 The effect of observational accuracy and implications of correlating to GMST

In this study, we used the txge25 indicator to quantify the warm summer of 2018. This is very similar to the more common indicator su (often referred to as summer days), defined as the number of days when $T_{max} > 25°C$. For model data, where the number of reported decimals are plenty, this choice has little to no consequence. However, for observations, and specifically from manual historical records, there is a notable difference between counting days when $tasmax > 25°C$ and where $tasmax \geq 25°C$. The reason is that decimal points were not prioritised in early observational practices, e.g. a thermometer displaying $25.4°C$ was likely recorded as $25°C$. In our case, the station averaged median FAR for the summer of 2018 event decreased from $\sim 0.65$ for the su index to $\sim 0.48$ for the txge25 index, an indication that using the former index results in an underestimation of warm days in the pre-industrial period.

The regression between rx1day (GridClim) and GMST (1989-2018) is relatively strong along, and in proximity to, the coast between 60 and 62°N (Fig. 8). Outside this sub-area, the regression is generally weaker and not statistically significant, with no distinguishable spatial patterns. In comparison, the regression between txge25 and GMST exhibits relatively small spatial variations (Fig. A1) over the domain, albeit with fewer significant grid points.

For the summer of 2018, shifting the distributions based on observations of the current period leads to an underestimation of FAR compared to the observation-only FAR for most of the stations (Fig. A7). Likely, this is a result of the weak regression coefficient between the variable (txge25) and GMST (Fig. A2). There are two stations, Karlshamn and Linköping-Malmslätt, where shifting the current distributions results in a higher estimation of FAR compared to the observations. This anomalous behaviour further supports that these stations suffer from inhomogeneities. For the Gävle event, differences between FAR based on scaling observations of the current period and FAR based on only observations are not as uniform (Fig. A8). At a few stations scaling the distributions of the current period results in a higher FAR compared to observations (e.g. Delsbo A, Härnösand), whereas the scaling results in a lower FAR at e.g. Gävle-Åbyggeby and Uppsala.

## 4 Conclusions

We have conducted two sets of attribution analysis on two notable extreme weather events in Sweden: The warm summer of 2018 and the heavy precipitation event in Gävle 2021. For the initial analysis we made use of a number of gridded datasets and assumed that the variable describing the event either shifted, or scaled, with GMST. This allowed us to calculate the probabilities, and their change, for the events during periods that represent the climate in a pre-industrial period and during the recent past. In the second analysis, we relied solely on observations to represent the climate during the pre-industrial (1882-1911) and current (1989-2018) periods to retrieve corresponding probabilities. For the summer of 2018, results using shifting by the relationship to GMST generally agree with the results based on observations when including all stations. However, the adjusted station average, excluding stations likely influenced by inhomogeneities, exhibits a stronger attribution compared to

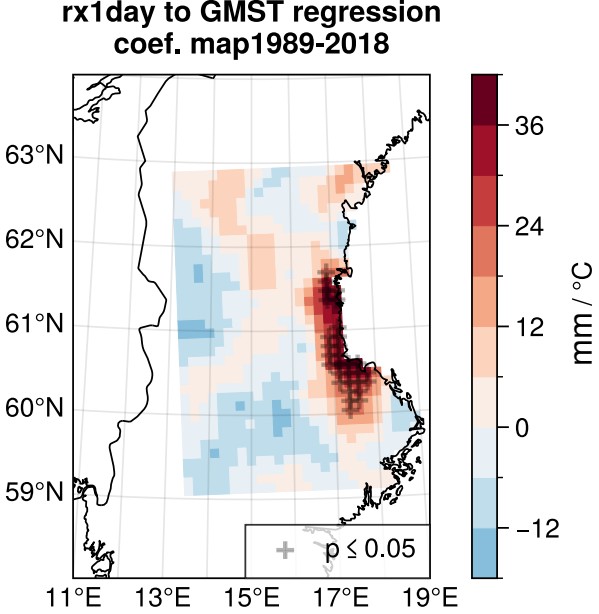

**Figure 8.** Map of the coefficients of the rx1day to GMST regression for GridClim. Crosses indicate significance at the $p \leq 0.05$ level.

the results of the shifted analysis. We also note that with homogenised observations the station-based FAR is likely to increase compared to the current results which are based on unhomogenised data, as the historical data tends to overestimate high temperatures. Overall, for the case of the summer of 2018, the proposed statistical method from Philip et al. (2020) agrees with the observation-based estimate. Furthermore, based on these results we can conclude that 1 out of 2 of every heat wave similar to the summer of 2018 can be attributed to changes in the climate. Alternatively, such heatwaves have become twice as likely due to changes in the climate. When only using station data, the previous statement increases to more than 2 out of 3, and would likely be even higher using homogenised data. Hence, there is a risk that studies that investigate temperature related events, relying solely on shifting following the GMST, underestimate the strength of the attribution.

For the precipitation event in Gävle 2021, our results show a fairly good agreement among the gridded datasets, with a median FAR $\sim 0.88$ and 5th percentile $> 0.5$, but the variation is large between the stations. A few stations exhibit FAR $> 0.5$, which is comparable to the gridded analysis, and it is only Gävle-Åbyggeby and Härnösand that exhibit a FAR that is significantly above 0. This uncertainty makes it difficult to derive any attribution statements about the precipitation event.

Comparing the two events, the study indicates that precipitation events appears to be more sensitive to choice of domain. The regression between temperature and GMST is also more spatially consistent compared to that between precipitation and GMST. With the high availability of observations in Sweden, a potential improvement to the station-based attribution is to use homogenised data. This would clarify the uncertainties that arise with using non-homogenised observations. However, this is mostly relevant to the investigations of extreme events where long-running observational networks are available. Regarding the



more generally applicable regression-based method for attribution, future studies should continue to explore how the regional variations in relationships, such as the that between local CC-scaling and GMST, affect the outcome of studies on extreme event attribution.

300   *Code availability.* Code used in this analysis is available at https://github.com/Holmgren825/holmgren_kjellstrom_exploring_attribution.





**Appendix A: Figures**

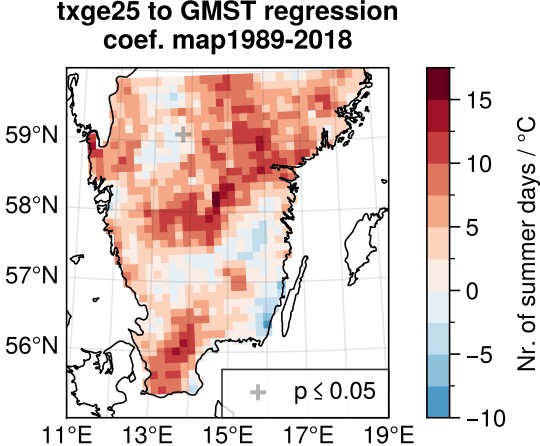

**Figure A1.** Map of the regression coefficients ($\beta$) of txge25 to GMST for GridClim. The cross indicates significance at the $p \leq 0.05$ level.

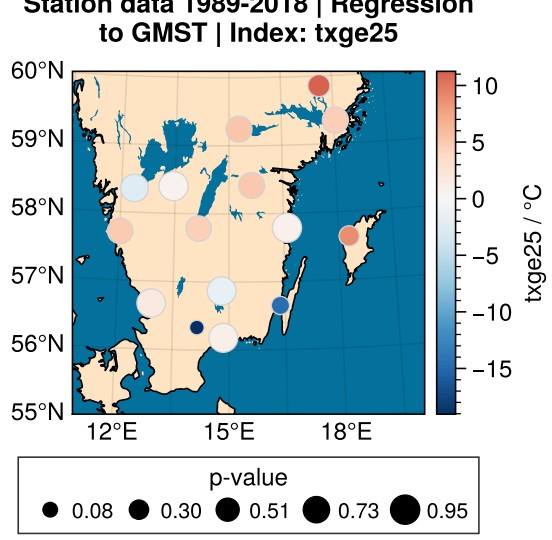

**Figure A2.** Regression between the index series and GMST at the respective stations. The strength, and sign, of the regression coefficient is indicated by the colour, while the size of the markers indicate the p-value.





**Figure A3.** Trend analysis for the txge25 index. KPSS ≤ 0.05 indicates that a series is non-stationary. ADF ≤ 0.05 indicates that a series is trend stationary.







**Figure A4.** Trend analysis for the txge25 index. KPSS $\leq 0.05$ indicates that a series is non-stationary. ADF $\leq 0.05$ indicates that a series is trend stationary.





**Figure A5.** Trend analysis for the rx1day index. KPSS $\leq 0.05$ indicates that a series is non-stationary. ADF $\leq 0.05$ indicates that a series is trend stationary.



**Figure A6.** Trend analysis for the rx1day index. KPSS $\leq 0.05$ indicates that a series is non-stationary. ADF $\leq 0.05$ indicates that a series is trend stationary.





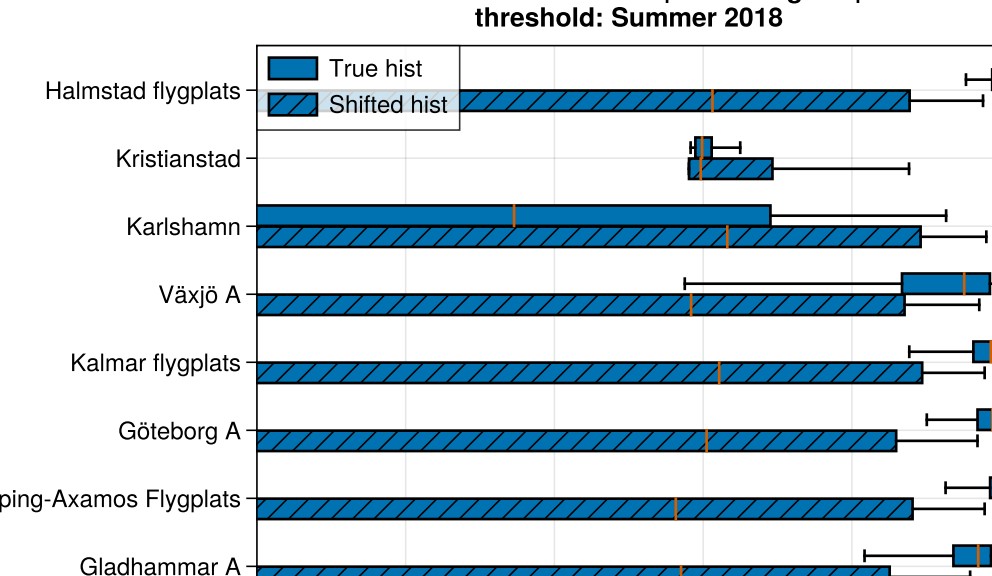

**Figure A7.** Comparison of two sets of FAR distributions. The first of the two bars in every pair is based on the true historic period, equivalent to what is shown in Fig. 5. The second bar shows the results from shifting the distribution of the current climate according to its regression to GMST.





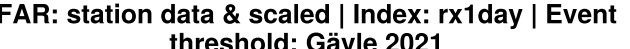

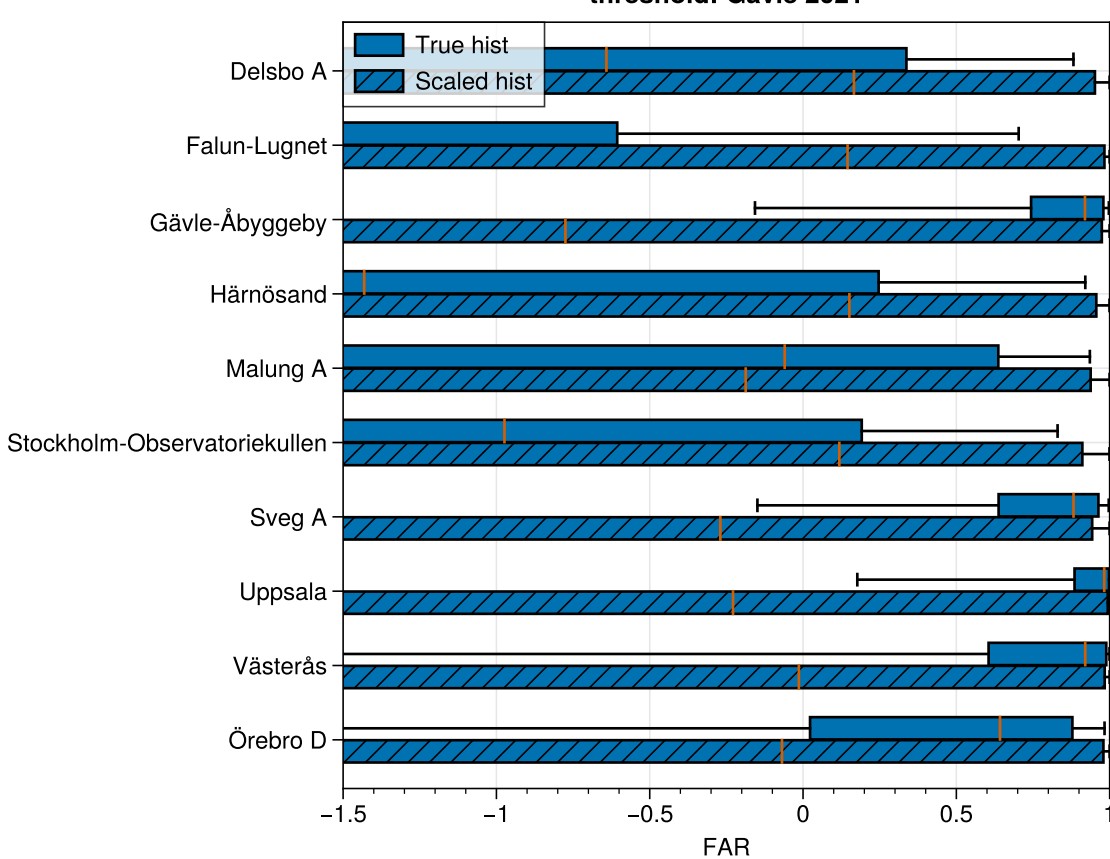

**Figure A8.** Comparison of two sets of FAR distributions. The first of the two bars in every pair is based on the true historic period, equivalent to what is shown in Fig. 7. The second bar shows the results from scaling the distribution of the current climate according to its regression to GMST.



*Author contributions.* EH developed and carried out the analysis, prepared figures, and wrote the article. EK initiated the study, provided comments during the development of the analysis, and assisted in the revision of the article.

*Competing interests.* The authors have no competing interests to declare.

305   *Acknowledgements.* This research was funded by the Swedish Meteorological and Hydrological Institute, specifically the 1:10 government grant "Klimatanpassning".



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
