# Peer review of "Exploring extreme event attribution by using long-running meteorological observations"

_EGUsphere, 2023_

## Author Comment (AC1)

**Response to referee 1**
https://doi.org/10.5194/egusphere-2023-2879-RC1

We thank referee 1 for taking the time to assess our manuscript and providing valuable feedback. In this document, we have included said feedback along with our responses in blue.

Exploring extreme event attribution by using long-running meteorological observations

This paper assesses two methods of event attribution for two events in Sweden – a hot summer and an intense rainfall event. It is shown that the two different methods agree reasonably for the temperature event, but show more disagreement for rainfall.

I found the introduction and methods clearly written and enjoyable to read, with good background literature (although some perhaps less relevant to this particular study). The methods are clearly explained and the results from the second method well presented – but I struggled to identify the results from the first method, or a clear comparison between methods.

**1   General comments**

Clearly labelling the two methods from the outset would be useful – the header of 2.2 is misleading, as observations are also used in the first method. Something like 'GMST adjusted method' and 'using pre-industrial observations' would be more accurate.

This is a good point, and we agree that the header of 2.2 should be more specific, and including pre-industrial observations in the heading is a good suggestion.

In order to provide a comparison of the two methods a more thorough presentation of the results of the GMST adjusted method is needed. The GMST adjusted method could be applied to the same datasets as used in the pre-industrial observations data – I am not clear if it is.

This is a very important point. We will extend the presentation of the GMST adjusted results to better match the extent at which the results from the pre-industrial method are presented. In the revised version, we will also update figure A7 and A8 following comments from referee 2 and move these to the main results. These compare the results from the method using pre-industrial data and the results from applying GMST shifting/scaling to the current period data, for each station.

The study presents the use of pre-industrial data for attribution as a good alternative to the GMST adjusted method, without full discussion of possible problems with the method. Greater emphasis on possible downfall of the pre-industrial observational data would be useful. One major advantage of using a shorter observational record with GMST is that the data needed is available for more locations and variable globally. Although long observational records are available in Sweden, there are many parts of the world where this is not the case – this should be better highlighted.

This is absolutely correct. But we don't aim to present the use of pre-industrial data as a suitable alternative to the GMST adjusted method. This study is about comparing/evaluating the GMST method to only observations. But this is an indication that we should try to be more clear about this. We have one or two sentences regarding the shortcomings of relying solely on pre-industrial data in the conclusion, but we agree this is a bit short, so we will elaborate on this.

**2 Specific comments**

Title – doesn't capture the content, too vague

We will reconsider the title.

Abstract - 'analogue approach' this term is widely used for a different method using dynamical analogues (e.g. Climameter). Perhaps adding 'statistical' would make it more clear what you are doing (see also comment above about labelling the two methods).

This is a good suggestion. Both could be considered statistical, but as mentioned above, we will change the description of the methods to something more aligned with your suggestion of GMST adjusted method compare to pre-industrial observations.

Paragraph at line 40 could come sooner in the introduction (around line 23) as the two paragraphs either side flow better together (and have some repetition in the local-global responses).

Good suggestion, thanks.

Lines 25-30 perhaps irrelevant to this study as physical processes are not covered in this statistical assessment.

We have included this since we think it is valuable to give a brief overview of the different approaches to attribution.

Line 18 – what types of events?

We will expand this sentence to include the specific examples from the cited references.

Fig1 – I find this a little unclear, is p0 more likely hot than p1?

In this hypothetical case of heatwaves (as measured by days with Tmax $\geq 25°C$), the probability p1 is larger than p0, so it is more likely to encounter a usually long heatwave

in the world of p1. However, we realise that the explanation of this figure could be more specific to make it clear what we want to show.

To include or not the event in question?
Not sure to what this comment refers. But in general, the question of including, or not, the event under investigation in an attribution study is relevant. We haven't touched upon this in the manuscript, but will add some sentences on this.

Why days greater than 25, not just Tmax?
The summer of 2018 in Sweden does not stand out when it comes to Tmax: there are plenty of years with a similar maximum temperature (this is just a single day after all). However, when it comes to the number of days exceeding 25°C, 2018 is essentially unprecedented. We will elaborate on this in the section about climate indicators.

Fig.3 caption typo in dates (1882-1992) - and fig 6
Thanks!

Fig.6 - no stations have at least one year missing 15% of days (none have the cross)? Is that correct?
This is correct, and we will remove the explanation of the cross to avoid confusion.

e.g. line 160, figA3, Interchanging use of historical and pre-industrial for the 1882-1911 period – I think it would be clearer to use pre-industrial throughout as historical could mean any past period (I think sometimes you use it to refer to the full historical/observational record).
Good catch. We agree and will make sure that we only use pre-industrial throughout the paper.

Paragraph at line 88 could be shortened, as the methods described are not those used in this study – perhaps it would be better to start with paragraph at line 97 stating what is done in this study, then mention that there are other methods used elsewhere.
We think that the information in this paragraph is valuable as it provides additional background as to why the GMST shifting/scaling is used. But since it is only background, we will condense it.

Line 112 – data for this study / event definition, a subheader would be useful here.
Good suggestion.

Header 2.2 - observations are used in the first method too
Agree, see reply for the general comment which also concerns this.

Section 2.3 - the climatic indicators have already been mentioned in the section above, maybe this should go into an event definition section – which perhaps could be section 2.1, before the two methods.

This is also a good suggestion.

---

## Author Comment (AC2)

**Response to referee 2**

https://doi.org/10.5194/egusphere-2023-2879-RC2

We thank referee 2 for taking the time to assess our manuscript and providing valuable feedback. In this document, we have included said feedback along with our responses in blue.

For two classes of weather events – the frequency of extremely hot days and the maximum 1-day precipitation accumulation - this paper compares the estimated fraction of risk attributable (FAR) to climate change using two approaches to event attribution: first by estimating exceedance probabilities in 30-year time slices representing the factual and counterfactual climates, which are assumed to be stationary, at individual stations; and then using a nonstationary trend fitted to a spatial average computed from gridded data products. Both methods are found to produce relatively similar results for the FAR of the number of very hot days, while the FAR for extreme precipitation is found to be somewhat variable, particularly in the station data.

The paper is clearly written, and the discussion around potential homogeneity issues in the station observations is a useful and important one. However, it's not entirely clear to me what the purpose of the comparison is here, or what the overall conclusions should be. This is perhaps because one method is used with station data, and another with gridded data, so it's hard to understand whether differences in the results arise from the dataset or the method used: I think this could be a really useful comparison if both methods were used with both station and gridded data.

**1 General comments**

The nonstationary method used here seems to only use 30 years of recent data to estimate the covariate β describing the strength of the relationship between the extreme and GMST (lines 114-115). This is a very short time series: usually in WWA studies we would use as much data as possible to estimate this parameter, partly because a large sample size is usually needed to get stable estimates of the model parameters and partly to reduce the risk of conflating the GMST trend with decadal variability. I would suggest using longer time series to fit the trends, which would give a really useful and interesting comparison of whether the linear regression really captures the changes between the two snapshots. If that's not possible due to data availability, you should highlight that only 30 years of data were used to estimate the nonstationary model parameters, and discuss what the implications might be.

*This is an important point, and we understand the need to further explain why we've chosen to compute the regression over the 30-year periods for the gridded data. As is also mentioned in the response to the specific comment concerning this topic (Figure A7/A8), we will update parts of the analysis of the station data. Specifically, we will extend the period over which the regression to GMST is computed before shifting/scaling the distribution based on station data of the current period.*

A GEV or Gumbel distribution is used to model block maxima/minima: there's no theoretical basis on which to use them to model txge25, which is a count variable. To simplify the statistical modelling, I'd suggest looking at maximum temperatures instead; if that's not feasible, you could try fitting a nonstationary Gaussian distribution to the log of the counts.

*You are correct in that there is no theoretical basis to model count data with GEV or Gumbel. We've employed a Kolmogorov-Smirnov test (KS-test) to make sure that the distributions represent the underlying data. Regarding the index, we have opted to use the txge25 index since this (much) better captures the severity of the summer of 2018 in Sweden. For an index like Tmax, a summer of 2018 is not unlikely, but in terms of txge25 it is essentially unprecedented. We will expand our reasoning on the event definition in the section on climate indicators.*

My understanding is that, since both $p_0$ and $p_1$ are nonnegative, the FAR can never be greater than 1 (equation 2): however, in both Figures 5 and 7 it looks as though FARs above 1 occur, although the axes are truncated at 1 so it's hard to see. Please check this, and also modify the axes so that the upper bounds of the confidence intervals are visible.

*Your understanding is correct. FAR above one does not occur in the data making up the figures, but we agree that it can seem this way due to the truncation. It is also worth noting that these distributions are very skewed in some cases. We will update the figures and extend the x-axis to make this clear.*

**2 Specific comments**

Abstract: I find the terminology here a little vague: it's not clear what is meant by 'the reference method' and, since both methods use obervations of some sort, this doesn't help to understand which is which. It would be useful to add a line explaining that the 'widely adopted' method uses a transient/nonstationary model, and rather than referring to the 'analogue approach' (which is becoming synonymous with another method), I would perhaps refer to a factual/counterfactual comparison.

*We will update how we describe the methods to be more precise.*

46, 52: The reader doesn't know what 'the reference method' is yet, or 'shifting and scaling' – this needs some introduction.

*Good catch. We will change this.*

62-63 & 67-68: The rapid attribution method could also be used on the long-running meteorological observations, so I think it would be useful to distinguish more clearly between the two methods: maybe 'we will also perform an analysis based on directly comparing the current and preindustrial periods in data from several stations with long observational records'. Also some repetition here, so 61-63 could be removed altogether.

This is a good suggestion.

77: This should be more precisely defined: $p_1$ and $p_0$ are the probabilities of observing an event of equal or greater magnitude than some threshold value in the factual (current) and counterfactual (preindistrial) climates ('exceedance probabilities').

We agree, thanks.

80. I found this a bit unclear – maybe 'FAR describes the proportion of events of the same (or greater) magnitude that can be attributed to the forced change'?

Also a good suggestion, thanks.

84. Change to 'The exceedance probability'

Will do.

88-96. I don't think I've seen examples of climate models being used to estimate $p_0$, although they are certainly used to estimate probability ratios. I'd suggest moving this paragraph to the description of the datasets

This is about explaining the background as to why the current methods, for instance GMST shifting, are used. One could, for example, use the pre-industrial control run of a GCM as a historical snapshot, similar to how we are using station data in this study, to remove the dependence of the regression to GMST.

101-2. Not all distributions have these three parameters: to make this more general, I'd remove this line and simply say that 'the mean $\mu$' is shifted following...

This is a good point.

105. '$\mu$ and the standard deviation $\sigma$ are...'

Thanks.

112-128. This breaks up the flow a bit – I'd move this (and maybe 88-96) into a separate subsection on datasets.

Good suggestion. We will move this to a new subsection.

Figure 1. This was quite hard to read in black & white, could you change the colour scheme to something more colourblind-friendly?

We will update this figure so that the lines representing the factual and counterfactual worlds also use different line styles, in addition to different colours.

129-130. The WWA approach outlined in Philip et al. (2020) uses maximum likelihood

estimation to estimate the parameters of a nonstationary GEV distribution directly from (3-5), rather than first fitting a linear regression to estimate the trend and then estimating the parameters of a stationary GEV separately. I wouldn't expect this to make much difference to the overall conclusions but this should be checked and commented on – you can fit the nonstationary GEV distributions using the online Climate Explorer tool provided by KNMI (first upload the time series, then choose the 'trends in return times of extremes' option).

These are very valuable insights, especially since it touches upon something we were unable to deduce from Philip et al. (2020). We will add a comment on this in the paper.

It would also be useful to be clearer about which time period was used for the regression and parameter estimation – and, if only 30 years is used, this would be a good opportunity to discuss the implications of using a relatively short time series.

Since this connected to one of the general comments, we refer to that one for the reply. But in general, we will elaborate on our reasoning behind the choice of time periods.

131. How was this 95% interval estimated?

The regression to GMST is computed using the python package statsmodels. The returned confidence interval is based on the Student's t-distribution.

141-2. Does this mean that the spread of all members was used to determine the confidence bounds? How was this done – was a parametric distribution used, or order statistics?

Yes, for CORDEX we used the spread from all ensemble members with an acceptable trend (see previous comment). We computed the quantiles directly (order statistics) on the resulting FAR ensemble.

158-9. Why was stationarity checked, and over which period? I can see the advantage of checking that each of the time periods studied could be treated as locally stationary, but as written, this could be read as suggesting that the full series was found to be stationary.

Here it is the two separate time periods that are checked for stationarity, and we should be more clear on this. This is used in the reasoning around if the data should be detrended or not (L.245-246).

175-7. You could add a line to explain why this is: the GEV with negative shape parameter has a finite upper bound, which can lead to observed events becoming theoretically impossibly in the shifted/scaled distribution. The Gumbel distribution, which has its shape parameter fixed at zero, has no upper limit and so does not exhibit this behaviour.

This is very useful. Thank you.

178-9. As noted above, a Gumbel distribution isn't theoretically justified for count data like txge25.

Since this is part of a general comment, we refer to that one for our reply.

189. Please add a line interpreting this FAR in terms of the number of hot days.

Will do.

191. How are percentiles of the FAR computed? Also, this notation is slightly confusing, because $p_0$ and $p_1$ have already been used to denote exceedance probabilities. Percentiles could perhaps be relabelled as $Q_5$.

We agree that the notation can be confusing here, and relabel it as $Q_5$ is a good suggestion. Percentiles/quantiles are computed directly (order statistics) on the 1000 FAR values resulting from the bootstrap.

Figure 4. I would expect the size of the circles to represent a range of values - it's not clear exactly what they refer to here.

The size of the circles here represent the range between the 95th and 5th percentile of FAR, essentially the uncertainty. We will improve the explanation of this in the figure caption.

213-4. Why are $P_5$ and $P_{25}$ given here, rather than an upper and lower limit?

For an attribution study, we reason that the lower limits of the confidence interval are more interesting, it is here we determine if the event can be attributed or not. But we agree that presenting these and at the same time discussing the spread is confusing, so we will add the upper limits of the confidence interval as well.

Figure 5 & 7. I don't understand how the FAR is greater than one in some of these cases – perhaps some additional scaling has been applied? Please extend the x-axis to show the upper bounds of the confidence intervals. It would also be useful to add a vertical line at 0, highlighting the critical threshold for evidence of an effect.

See the reply to the general comment on the same topic.

236-240. You could also discuss the fact that observations of precipitation are typically more variable than observations of temperatures; and that gridded data, by its very nature, will not tend to contain such extreme extreme values as a single station, which may result in a better constrained distribution. When trying to fit a distribution to only 30 years of data we don't really expect to get an accurate estimate of the return level of any events with a return period of greater than 30 years (or, conversely, an accurate estimate of the return period of particularly extreme events): this may also lead to inflated estimates of the return period, which can in turn make the PR and FAR estimates unstable.

This is very valuable input. We will extend this discussion to include a part on the variability of precipitation and how it is not necessarily represented in gridded data.

251-258. I think this could fit better in section 2.3, where the climate indicators are introduced.

We will move this to 2.3

259-262. This discussion of spatial variability in the trend is really interesting and could be referred to in the discussion of variation between the stations – I'd like to see Figure A1 in the main text, perhaps with the station regression coefficients overlaid so that the similarities/differences between the gridded product and the stations are really clear.

This is a good suggestion. We will try to combine Figure A1 and A2 and see how this could fit into the main text.

292. You could mention that attribution of extreme precipitation events is known to be sensitive to the event definition, both in terms of the spatial domain and the duration of the event.

Agree.

294-5. I think that most studies would try to use homogenised data, where available: you could frame this instead as highlighting the importance of using homogenised data.

Good point.

296-299. The conclusions concerning the two different methods are rather weak, perhaps because it was never very clear what the purpose of the comparison actually is. Gridded datasets offer an invaluable opportunity to examine spatial variability in trends and FAR over a whole region, but should be validated against station data if possible to ensure that they are locally accurate. However, it's hard to get a sense of their relative merits here because two different methods have also been used, so there's very little common ground for comparison.

We completely agree that gridded datasets are an invaluable asset when it comes to attribution studies. The purpose of this study is not to evaluate the use of gridded data, but the evaluation of retrieving FAR/PR by shifting/scaling a distribution of "current" data according to its regression to GMST. To achieve this, we wanted to retrieve FAR while staying as close to the data as possible (e.g. avoiding regressions), and compare this to the traditional approach. Since generally, gridded data before ~1960 is uncommon, and long-running observations are available, we opted to use these. So it is not possible to apply the method we've used on the station data to the gridded data, since these don't provide the pre-industrial snapshot. Furthermore, we will improve how the GMST shifted method is applied to station data, see reply to the comment about figure A7/A8. Overall, based on this feedback, we realise that we need to clarify that the purpose of the study is to evaluate/explore the GMST shifted method, not the use of gridded data.

299. CC-scaling has not yet been defined.

Good catch, thanks.

Figure A7/A8. I don't quite understand what these figures show. Is it the case that the upper bar shows the FAR computed from $p_0$ and $p_1$ computed from stationary distributions corresponding to the historical and current periods; while the second bar (shaded) shows the FAR based on a linear regression estimated over the 'current' climate only?

Given that the shorter time periods have been tested for stationarity, and no trend signal could be detected, it's not surprising that the confidence intervals of the hatched bars all include zero. A fairer and more useful comparison would be to estimate the regression coeffiecients over the whole period, to see whether the regression model adequately captures the observed difference between the two 30-year slices.

You seem to be interpreting the figures correctly. The upper bar of a pair shows the FAR based on current and pre-industrial data, using no regression to GMST. The lower bar of a pair shows the FAR computed using only the current period, relying on the shifted/scaled distribution to compute $p_0$. Your point about computing the regression over only the stationary period is very valuable, and we will remake the analysis behind this plot and make use of a longer period for calculating the regression.

---

## Referee Report (RR1)

**2023-2879-ATC1 – reviewer comments**

The authors have fully accommodated the suggestions made in the first round of reviews, and I think the paper is much stronger as a result. In particular, the comparison between the two methods using the same data is very illuminating, suggesting that the relationship between local and global temperatures may (at least in come cases) be well modelled using only a relatively short time series. There are still a few points that could be clearer, and I've highlighted these below. Generally these are fairly minor queries, but toward the end of the paper I became concerned that I may have misunderstood how the WWA-method is applied (see my comment on Figure A2), in which case more substantial revision of that section may be required to fully clarify what was actually done and avoid any potential for misinterpretation.

**Specific comments**

18-19. I think 'IPCC 2021' is redundant here – unless there's another chapter that could be cited specifically?

44. Remove added 'had'

50-54. I don't think this describes the WWA method very clearly – in particular, it's not clear that this approach is used for trend fitting (I initially took line 51 to mean a comparison of conditions in two distinct periods!). I'd suggest something like 'Instead, it is possible to build a statistical model in which the distribution of the variable(s) describing the event changes with global mean surface temperature (GMST), and to use this to estimate the magnitude of events in the preindustrial climate'.

67. WWA studies often have found a fairly linear response, particularly for heat extremes, so 'is unlikely to capture' seems a bit strong – suggest 'may not always adequately capture'

83, 88. Unnecessary to redefine WWA-method after line 55.

97. Strange formatting, line starts with a comma

101. It's not clear from the text how the models were evaluated – please briefly describe the four metrics used, and direct the reader to where in the text the results of the model evaluation are reported.

108-119. A minor comment, but this section would be easier to follow if you defined the 2018 heat event, then the 2021 rainfall, rather than flipping between the two as written.

141. It would be useful to define 'exceedance probability', maybe in line 133 – it may not be clear to all readers that you're still talking about $p\_0$ and $p\_1$.

145-159. I don't think this discussion of how to estimate $p\_0$ belongs in the 'WWA attribution' section – maybe it could be moved to the introduction, around line 49.

191. This description of the WWA method could still be clearer – suggest 'they estimate \beta, \mu and \sigma, along with any other model parameters, directly from equ. 3-5, and would use the longest available reliable time series rather than only a subset as we use here'.

194-197. This section would be easier to follow if 'For all datasets, we used the regression coefficients… equations 3, 4 and 5' were moved to line 189, directly after 'time series of each index'.

211. Change 'a station' to 'any stations'

234-245. This would fit better in the section on event definitions.

254. The GEV could also have a positive shape parameter – suggest 'the GEV distribution has a finite upper bound when the shape parameter is negative'.

258. Please add a line addressing the potential criticism that the Gumbel/GEV is theoretically justified only for block maxima and not for count data.

292. I found this a bit hard to interpret – does this mean that temperatures were overestimated in the earlier part of the record? Please clarify in the text.

297. Change to 'FAR similar to, albeit lower than'

309. Maximums → maxima

Fig 5 & 7 caption: change to 'The bar represents the percentiles and median of the…' 'Green bars denote the average for each method (PI and WWA)'.

Figures 5 & 7. Does the WWA average include the CORDEX runs? From the text and plot it's not clear. If not, perhaps move the CORDEX bar below the WWA average, to make this absolutely clear.

331. I think it's still right to say that uncertainties are higher for the stations, rather than for the PI method – suggest changing this back

336. The 30-year period limits how well any time series can represent variability, especially when evaluating an event that may not occur within that 30-year period. I'm also not sure what is meant by 'over constrained distribution' here. I'd say instead that this may lead to unstable estimates of the return period/return level, and hence to unreliable estimates of FAR.

353. Again, it's not totally clear from the text what was actually done here. The PI-method has already been applied to the long-term observations; as I understand it, you now use the WWA-method to estimate the FAR using a trend fitted to only the current (1992-2021) climate, and compare the results. If this is correct, please update these lines accordingly.

361-362. While it is true that most of the regression coefficients seem to be close to zero, the northernmost point seems to be close to 10, but is still non-significant. Might this therefore also be because of high variability, as well as a weak mean trend?

366-367. I think you could go further with this conclusion: this result implies that the long-term relationship between GMST and local temperature extremes can be estimated using only a relatively short time series from the recent past, which in turn implies that the relationship between GMST and local temperature extremes – at least, in this area and for these extremes – remains fairly constant.

373. I'm still not sure what is meant by 'over constrained' in this context, please explain.

375. Please add a description of Figure 10 and explain what it is showing. How was significance determined? I'm surprised that a temperature-based index has so few significant points, can you suggest why this might be?

Figure 10. The colour scale here is a bit confusing - please redraw so that increased rainfall is shown in blue (the BrBG colourmap in Python would be useful to distinguish precipitation from temperature results)

385. One point that would be worth investigating is that the WWA approach assumes that the scale of the temperature distribution remains constant over time – Figure 3 suggests that this may not be a realistic assumption, because the PI temperature distributions are typically somewhat narrower than the 1991-2021 distributions. This would mean that the WWA-PI distributions are typically wider than their PI-PI counterparts, and that the event is therefore deemed more likely in PI than it really was – which would lead to the lower FAR seen in Figure 8. The effect is less clear in the precipitation series, and the situation is less straightforward because the scale parameter changes with the location: so it's not clear whether the scaled distribution would be likely to systematically misrepresent the PI distribution. However, this could be easily checked by comparing the scaled distributions to Figure 3.

393. I think you mean exceedance probabilities here?

406. Or possibly the fixed scale parameter – which could be allowed to vary

406. The differences between the two results are fairly minor, as you highlight around lines 364-367: this should be restated in the conclusion, along with a summary of my comment above about lines 366-367.

408. One potential advantage of the WWA method is that, if homogenised station data aren't available, it could be modified to accommodate a changepoint in the series and so to make use of all of the available data to estimate the effect of GMST.

419. **Gridded** datasets (worth highlighting: to distinguish from the station dataset you mention in the next line)

423-425. It's not clear where in the main text this is indicated: is this a reference to Figure 10? If so, you should also draw this conclusion where the figure is discussed, perhaps at line 380. However, the next line suggests not. Please clarify.

432. You should also highlight here that WWA recommend using as much data as possible to estimate the model parameters, almost certainly more than 30 years of data. Although the mean trend in temperatures was fairly well estimated, the uncertainty probably was not: using a longer time series would be expected to give a better estimate of the variance of the distribution, which is critical for correctly estimating return periods and PR/FAR (see eg. Zeder et al. 2023, 'the effect of a short observational record on the statistics of temperature extremes')

Figure A2. At this point, I started to doubt whether I've misunderstood something in the methods. My understanding is that the WWA-method was implemented for station data by regressing these time series against GMST, and using Gumbel distributions corresponding to present-day and pre-industrial GMST to estimate the FAR. The tests reported suggest that there is no evidence of nonstationarity, but the estimated FAR is consistently strongly positive, which implies a

nonstationary trend. How do you reconcile these two findings? (If I've misunderstood the methodology then that section needs to be rewritten to clarify what was actually done.)

Figure A2 & A3/A4 & A5 both have the same caption – please update to clarify that one is for the current period, and one historical.

---

## Author Response (AR2)

**Author's response**

**1   Referee 1**

We thank referee 1 for taking the time to re-assess our manuscript and providing valuable feedback.

The authors have taken on board both mine and the other reviewers comments - and I think the new manuscript is substantially improved. I am glad to see a renaming of the methods which makes it much easier to follow. Further clarifications in the methods section are useful.

I notice a minor typo in the brackets of line 92.

We can't find a typo on line 92 in either of the reviewed version of the manuscript or the pdf showing the tracked changes.

Only other comment is that the abstract contains references - this may not be allowed.

We removed the citation, but kept the footnote to WWA.

**2   Referee 2**

We thank referee 2 for taking the time to re-assess our manuscript and providing valuable feedback.

The authors have fully accommodated the suggestions made in the first round of reviews, and I think the paper is much stronger as a result. In particular, the comparison between the two methods using the same data is very illuminating, suggesting that the relationship between local and global temperatures may (at least in come cases) be well modelled using only a relatively short time series. There are still a few points that could be clearer, and I've highlighted these below. Generally these are fairly minor queries, but toward the end of the paper I became concerned that I may have misunderstood how the WWA-method is applied (see my comment on Figure A2), in which case more substantial revision of that section may be required to fully clarify what was actually done and avoid any potential for misinterpretation.

**2.1 Specific comments**

18-19. I think 'IPCC 2021' is redundant here – unless there's another chapter that could be cited specifically?
  Fixed

44. Remove added 'had'
  Fixed

50-54. I don't think this describes the WWA method very clearly – in particular, it's not clear that this approach is used for trend fitting (I initially took line 51 to mean a comparison of conditions in two distinct periods!). I'd suggest something like **'Instead, it is possible to build a statistical model in which the distribution of the variable(s) describing the event changes with global mean surface temperature (GMST), and to use this to estimate the magnitude of events in the preindustrial climate'**.
  Great suggestion, we have incorporated this.

67. WWA studies often have found a fairly linear response, particularly for heat extremes, so 'is unlikely to capture' seems a bit strong – suggest 'may not always adequately capture'
  Fixed, thanks!

83, 88. Unnecessary to redefine WWA-method after line 55.
  Fixed, thanks!

97. Strange formatting, line starts with a comma
  Only seems to be a problem in the diff/tracked changes.

101. It's not clear from the text how the models were evaluated – please briefly describe the four metrics used, and direct the reader to where in the text the results of the model evaluation are reported.
  We have added a short list of the metrics and hopefully made it clearer what the reference is.

108-119. A minor comment, but this section would be easier to follow if you defined the 2018 heat event, then the 2021 rainfall, rather than flipping between the two as written.
  This is a very good suggestion, we have re-organized this section.

141. It would be useful to define 'exceedance probability', maybe in line 133 – it may not be clear to all readers that you're still talking about $p_0$ and $p_1$.
  This has been added.

145-159. I don't think this discussion of how to estimate $p_0$ belongs in the 'WWA attribution' section – maybe it could be moved to the introduction, around line 49.

*This section goes into some details which requires knowledge of the method, and we think it would be tricky to include it in the introduction without extending this too much, borrowing what belongs to the method.*

191. This description of the WWA method could still be clearer – suggest 'they estimate $\beta$, $\mu$ and $\sigma$, along with any other model parameters, directly from equ. 3-5, and would use the longest available reliable time series rather than only a subset as we use here'.
*Fixed.*

194-197. This section would be easier to follow if 'For all datasets, we used the regression coefficients... equations 3, 4 and 5' were moved to line 189, directly after 'time series of each index'.
*This has been moved accordingly.*

211. Change 'a station' to 'any stations'
*Fixed.*

234-245. This would fit better in the section on event definitions.
*This has been moved.*

254. The GEV could also have a positive shape parameter – suggest 'the GEV distribution has a finite upper bound when the shape parameter is negative'.
*Thanks! We have incorporated this.*

258. Please add a line addressing the potential criticism that the Gumbel/GEV is theoretically justified only for block maxima and not for count data.
*This has been added.*

292. I found this a bit hard to interpret – does this mean that temperatures were overestimated in the earlier part of the record? Please clarify in the text.
*We have tried to clarify this.*

297. Change to 'FAR similar to, albeit lower than'
*Thanks!*

309. Maximums → maxima
*Thanks!*

Fig 5 & 7 caption: change to 'The bar represents the percentiles and median of the...' 'Green bars denote the average for each method (PI and WWA)'.
*Thanks!*

Figures 5 & 7. Does the WWA average include the CORDEX runs? From the text and plot it's not clear. If not, perhaps move the CORDEX bar below the WWA average, to

make this absolutely clear.

*It does. We have added a short passage that hopefully makes this clearer.*

331. I think it's still right to say that uncertainties are higher for the stations, rather than for the PI method – suggest changing this back

*We changed it so that it mentions both the PI-method and stations.*

336. The 30-year period limits how well any time series can represent variability, especially when evaluating an event that may not occur within that 30-year period. I'm also not sure what is meant by 'over constrained distribution' here. I'd say instead that this may lead to unstable estimates of the return period/return level, and hence to unreliable estimates of FAR.

*This is a good suggestion, we have changed the formulation of these sentences.*

353. Again, it's not totally clear from the text what was actually done here. The PI-method has already been applied to the long-term observations; as I understand it, you now use the WWA- method to estimate the FAR using a trend fitted to only the current (1992-2021) climate, and compare the results. If this is correct, please update these lines accordingly.

*We have tried to clarify what was done under 'Comparing the PI and ...'*

361-362. While it is true that most of the regression coefficients seem to be close to zero, the northernmost point seems to be close to 10, but is still non-significant. Might this therefore also be because of high variability, as well as a weak mean trend?

366-367. I think you could go further with this conclusion: this result implies that the long-term relationship between GMST and local temperature extremes can be estimated using only a relatively short time series from the recent past, which in turn implies that the relationship between GMST and local temperature extremes – at least, in this area and for these extremes – remains fairly constant.

*This is a good suggestion and we have added a sentence on this.*

373. I'm still not sure what is meant by 'over constrained' in this context, please explain.

*In line with the previous comment on a similar topic, we changed this to instead be framed as a lack of variability.*

375. Please add a description of Figure 10 and explain what it is showing. How was significance determined? I'm surprised that a temperature-based index has so few significant points, can you suggest why this might be?

*We have added two sentences to better explain figure 10. The regressions were computed using the ordinary least squares from the library statsmodels, here p-values are given by a t-test.*

Figure 10. The colour scale here is a bit confusing - please redraw so that increased

rainfall is shown in blue (the BrBG colourmap in Python would be useful to distinguish precipitation from temperature results)

*In general, when plotting precipitation, I would agree. However, for these figures, we think that it is the strength of the regression that is of importance. And keeping the colormap the same for the two plots makes this comparison easier.*

385. One point that would be worth investigating is that the WWA approach assumes that the scale of the temperature distribution remains constant over time – Figure 3 suggests that this may not be a realistic assumption, because the PI temperature distributions are typically somewhat narrower than the 1991-2021 distributions. This would mean that the WWA-PI distributions are typically wider than their PI-PI counterparts, and that the event is therefore deemed more likely in PI than it really was – which would lead to the lower FAR seen in Figure 8. The effect is less clear in the precipitation series, and the situation is less straightforward because the scale parameter changes with the location: so it's not clear whether the scaled distribution would be likely to systematically misrepresent the PI distribution. However, this could be easily checked by comparing the scaled distributions to Figure 3.

*This is an interesting comment, and as we interpret it touches on a somewhat larger topic: should the scale of the temperature distribution be kept constant or also be changed, as is done when scaling the precipitation distributions. As you mention, the current and pre-industrial observations seem to indicate that the scale does not remain. We consider a full investigation of this out of scope for this paper, but will add a few sentences on it in the results and conclusions.*

393. I think you mean exceedance probabilities here?

*Yes, fixed.*

406. Or possibly the fixed scale parameter – which could be allowed to vary

*Part of the comment to line 385 above.*

406. The differences between the two results are fairly minor, as you highlight around lines 364- 367: this should be restated in the conclusion, along with a summary of my comment above about lines 366-367.

*This has been adressed.*

408. One potential advantage of the WWA method is that, if homogenised station data aren't available, it could be modified to accommodate a changepoint in the series and so to make use of all of the available data to estimate the effect of GMST.

419. Gridded datasets (worth highlighting: to distinguish from the station dataset you mention in the next line)

*Agree, fixed.*

423-425. It's not clear where in the main text this is indicated: is this a reference to Figure

10? If so, you should also draw this conclusion where the figure is discussed, perhaps at line 380. However, the next line suggests not. Please clarify.

*This has been clarified in the relevant results section and in the conclusions.*

432. You should also highlight here that WWA recommend using as much data as possible to estimate the model parameters, almost certainly more than 30 years of data. Although the mean trend in temperatures was fairly well estimated, the uncertainty probably was not: using a longer time series would be expected to give a better estimate of the variance of the distribution, which is critical for correctly estimating return periods and PR/FAR (see eg. Zeder et al. 2023, 'the effect of a short observational record on the statistics of temperature extremes')

*Added a short section on amount of data.*

Figure A2. At this point, I started to doubt whether I've misunderstood something in the methods. My understanding is that the WWA-method was implemented for station data by regressing these time series against GMST, and using Gumbel distributions corresponding to present-day and pre- industrial GMST to estimate the FAR. The tests reported suggest that there is no evidence of nonstationarity, but the estimated FAR is consistently strongly positive, which implies a nonstationary trend. How do you reconcile these two findings? (If I've misunderstood the methodology then that section needs to be rewritten to clarify what was actually done.)

*Your understanding of how we implemented the WWA-method for station data sounds correct. As we interpret it, for the short snapshots, the trend test and regression to GMST are not directly comparable. The year-to-year variations within the trend stationary e.g. txge25 data can still follow the variations in the snapshot of the GMST data, which would give a positive regression. Furthermore, it is only the snapshots that appear trend stationary, if we were to do a similar test of the whole observational record, there likely would be evidence for non-stationarity. It is quite interesting that the regression between subsets of GMST and txge25 and the subsequent shifting according to the change in GMST can capture this.*

Figure A2 & A3/A4 & A5 both have the same caption – please update to clarify that one is for the current period, and one historical.

*Thanks, fixed.*